# Randomized Benchmarking of Local Zeroth-Order Optimizers for Variational Quantum Systems

## Abstract

In the field of quantum information, classical optimizers play an important role. From experimentalists optimizing their physical devices to theorists exploring variational quantum algorithms, many aspects of quantum information require the use of a classical optimizer. For this reason, there are many papers that benchmark the effectiveness of different optimizers for specific quantum learning tasks and choices of parameterized algorithms. However, for researchers exploring new algorithms or physical devices, the insights from these studies don't necessarily translate. To address this concern, we compare the performance of classical optimizers across a series of partially-randomized tasks to more broadly sample the space of quantum learning problems. We focus on local zeroth-order optimizers due to their generally favorable performance and query-efficiency on quantum systems. We discuss insights from these experiments that can help motivate future works to improve these optimizers for use on quantum systems.

## 1 Introduction

Quantum computing has over time gathered more and more attention from researchers for the promise of significant computational speedups relative to classical computers. This has spurred many developments across all fronts in the field, from algorithms to building real quantum computers. However, many of these works still rely on the use of classical optimizers. For instance, variational quantum algorithms are a class of algorithms that have parameters that are then optimized by a classical optimizer [Cerezo et al., 2021]. These algorithms exist both as a way to do machine learning in a quantum system and to realize practically useful algorithms in noisy near-term devices. And beyond the scope of algorithm design, classical optimizers also play a part in assisting experimentalists working in quantum information. Beyond simply being used in practice to realize theorized algorithms on real devices [Ebadi et al., 2022], optimizers can tune the control of physical actions (such as laser pulses, injections of electrical current, etc.) that all need to be controlled precisely to produce desired quantum operation [Coopmans et al., 2021, Leng et al., 2023].

For these reasons, understanding how classical optimizers interact with quantum objects is both difficult and highly important for designing the best quantum devices / algorithms. As a result, there are many studies on exactly that. Some of these studies are general benchmarks that compare a wide variety of optimizers [Pellow-Jarman et al., 2021, Anand et al., 2021, Singh et al., 2023]; others are works that propose new optimizers for quantum circuits and show experimental evidence for the advantages of their optimizer [Sung et al., 2020, Gacon et al., 2021, Leng et al., 2023]. However,

Submitted to NeurIPS 2021 AI for Science Workshop.

most of these works benchmark for a fixed set of problems using a specific form of parameterized quantum ansatzes / models intended for each problem.

While these works are important for understanding specific use cases, it's not necessarily clear how much the insight from these works translates to new scenarios. And in the case of physics experimentalists, similar concerns arise when there may be noise or aspects of their system that shift over time [Proctor et al., 2020, Blume-Kohout et al., 2020]. So unless they constantly re-evaluate many optimizers on their own, what sort of confidence can they have in their choice of optimizer being wise?

These are the questions we take a shot at addressing in this work. We do this by benchmarking on tasks that are randomized. So in addition to the random parameter initialization common in other benchmarks, we also randomize the parameterized circuit / ansatz that is used, and in some cases randomize parts of the objective we are trying to minimize. While this is certainly not the most perfect way to answer the questions we posed before, we hope that by adding more variety via randomness in our benchmark we can begin to identify features of optimization algorithms that work more generically on many types of variational quantum learning problems.

However, when doing a study like this, you run into the risk of benchmarking something so generic that it's difficult to get any concrete insights from results. For this reason, we narrow our focus onto understanding how to improve a specific class of optimizers. First, we only consider **zeroth-order** methods. As mentioned previously, this is because methods that only sample and don't require gradients currently tend to be more easily realizable on quantum systems. Second, we use only **local** optimizers. This means that our optimizers sample the objective centered around a specific "canidate" point. Last, we only consider **sample-efficient** methods. This means that our optimizers make optimization decisions based on sampling as few points as possible per step. These choices mostly centered around us deciding to focus on studying the SPSA algorithm [Spall, 1998] and optimizers like it, because its generally favorable performance and runtime efficiency on quantum systems.

In this study, we benchmark randomized experiments for a variety of Hamiltonain minimzation and generative modeling tasks. We side-by-side compare the performance of 7 optimizers: SPSA, AdamSPSA, 2-SPSA, QNSPSA, GES, xNES, and sNES. We produce plots illustrating both the average rate of convergence and statistics on the end-result performance of each optimizer, and discuss our thoughts on insights to be gained from these results. But in short, we believe there are two *main take-aways*. First, **more sophisticated optimizers are not generally better**. In our benchmarks SPSA tends to perform best overall, followed by the other simpler heuristic methods like AdamSPSA and GES. There is more nuance to this statement and it certainly isn't true in all circumstances, but under our randomized tasks methods simpler methods tended to be more reliably effective. Second, **there is a need for more robust or adaptive optimization heuristics**. While there are heurstics that can assist these optimizers in optimizing quickly in certain parts of the optimization process, at other points these heuristics can begin to hurt the optimizer's performance. As such, we argue that it would be beneficial to make these heuristics more robust to distribution shifts. But more broadly, we hope that this work helps stimulate thought into how to more generally compare and study optimizers of quantum systems.

## 2  Prior Work

There are a number of works that explicitly benchmark a variety of optimizers. Pellow-Jarman et al. [2021] compares a variety of both gradient and gradient-free optimizers on variational quantum linear solver problems, both in the presence and absense of noise. They show that while there's no clear best otpimizer, SPSA tends to perform favorably in realistic noise scenarios. Anand et al. [2021] benchmarks natural evolutionary strategies (NES) on variational quantum eigensolver (VQE) and state preparation problems. They also empirically investigate and provide some justification for how NES could be used in a hybrid algorithm to assist gradient-based optimizers in barren plateau

82 regimes. Singh et al. [2023] benchmarks optimizers for a variety of quantum chemistry tasks. Like
83 other studies, there's no clear best algorithm, but SPSA tends to perform well in noisy conditions.

84 Additionally, while not explicitly a benchmark, a number of works compare optimizers against
85 a variety of tasks. [Sung et al., 2020] introduces methods that use quadratic fitting of sampled
86 points to evaluate the gradient and perform gradient descent and policy gradient descent. They
87 additionally benchmark these methods against a variety of optimizers for three unique Hamiltonian-
88 minimization problems with specific ansatzes. They also include some more practical considerations,
89 such as the cost of evaluating different Hamiltonian measurments, the possibility of parallelizing
90 multiple quantum circuit evaluations, and doing more robust hyperparameter tuning. In their results
91 their method performs best, but SPSA can come close and often out-performs other methods in
92 success rates. [Gacon et al., 2021], which proposes the QNSPSA algorithm we use in this study,
93 also compares its performance to original SPSA on a variety of tasks and compares robustness to
94 parameter initialization. [Leng et al., 2023] does the same for their proposed AdamSPSA to SPSA
95 and similar finite-difference methods, but they instead compare on the task of tuning the performance
96 of a qubit operation on quantum computer.

## 3 Benchmarks

98 In this section we outline each of the benchmarks we perform in this paper, motivate the reasoning
99 behind each experiment choice, and provide the finer details of each. While we overall strive to
100 include some aspect of randomness / broadness, we make a few distinct choice of fixing specific
101 elements between different benchmarks. Some of these choices are just so we can help distinguish
102 any differences between types of learning problems, and others are so we try to understand differences
103 between different levels of difficulty within a type of learning problem.

104 To ensure fairness, for all experimental runs, all randomness (initialization parameters + random
105 circuits / Hamiltonians / distributions) are controlled by the random key. So although each run is
106 randomly sampled, because we use the same random keys across all optimizers, they all run the same
107 variations of each problem. The statistics of each run only vary as a result of the differences between
108 each optimizer.

109 We also want to ensure that our results aren't biased by a poor choice of hyperparameters. However,
110 especially because we are doing highly randomized tasks, it's difficult to identify what's a "good"
111 choice of hyperparameter means. And even if they could be identified, it's not always reasonable
112 to assume the user of said optimizer would be able to properly find them. Our compromise is to
113 do hyperparameter tuning only on a small subset of the possible problem space. We select the
114 hyperparameters according to a random search run on 3 random keys. This means that as we try
115 random hyperparmeter combinations, they will be tried on 3 different random configurations of the
116 learning problem. So while this is not as exhaustive as the 100 we test on, the tuning isn't heavily
117 biased to a single random problem sample. When there is a range of hyperparameters that all perform
118 optimally, we bias our choice towards the default values typically used by the algorithm's authors or
119 commonly selected in the literature. Once we have the tuned hyperparmeters, we benchmark each
120 optimizer on each problem using 100 runs. All models in all experiments have their parameters
121 initialized from a normal distribution of mean 0 and standard deviation $\pi$.

### 3.1 Hamiltonian Minimization Experiments

123 First we run experiments on Hamiltonain minimization problems. This means that we choose a
124 Hamiltonian as an observable, and the expected value of measuring this Hamiltonian becomes
125 the "loss" with which we are aiming to produce a quantum state that minimizes this loss. These
126 benchmarks are meant to encapsulate use-cases such as variational quantum eigensolvers and quantum
127 optimization problems that map some problem to a specific Hamiltonian. We produce our candidate
128 states by parameterizing a quantum circuit and optimizing it to map a simple state (usually $|0\rangle$) to the
129 state we measure with the Hamiltonian observable.

It is also important to note that we are simulating the noise-free version of this problem, as we assume we have access to the exact expected value of the Hamiltonian to minimize. While this is certainly not a realistic assumption, we wanted to first focus on how the aspects of each optimizer affect performance on quantum systems first before considering varying levels of noise as a factor.

All of the experiments we run here use random circuits as their quantum circuit / ansatz. Specifically we use the RandomLayers circuit by Pennylane [Bergholm et al., 2022], which are layers of randomly placed parameterized single qubit X, Y, or Z rotation gates mixed with randomly placed CNOT gates.

**1D Ising Model:**  The first set of experiments we run use the 1D Ising model as our Hamiltonian observable. Specifically, we use

$$\mathcal{H} = -\sum_{i=1}^{N} Z_i \otimes Z_{i+1} - \frac{1}{2}\sum_{i=1}^{N} X_i.$$

As the 1D ising model is known to be an easily solvable problem, the intention of these experiments is to provide a simple baseline to understand how our optimizers perform on easier quantum optimization problems. These experiments are run on systems of 3 qubits, with circuits of 30 parameterized single qubit gates and 10 2-qubit gates. This is relatively simpler and over-parameterized compared to the other experiments in this section. Each run is executed for 500 update steps of the optimizer.

**2D Heisenburg Model:**  Next we benchmark our optimizers on the 2D Heisenburg model. This serves as our harder problem, as not only do we include additional interaction terms w.r.t. the 1D Ising model, but now we also increase the dimensionality of the connectivity of our observable to a 2D lattice. We specifically use

$$\mathcal{H} = -\frac{1}{2}\sum_{i=1}^{N}\sum_{j=1}^{N}\sum_{M\in\{X,Y,Z\}} M_{i,j}\otimes M_{i+1,j} + M_{i,j}\otimes M_{i,j+1} - \frac{1}{4}\sum_{i=1}^{N}\sum_{j=1}^{N} Z_{i,j}.$$

In contrast to our 1D Ising experiments, these experiments are intended to gain insight into how our optimizers perform on a much more difficult problem. We use 9 qubit systems in these experiments, with quantum circuits containing 162 parameterized single qubit gates and 49 2-qubit gates. Each run is executed for 2000 update steps of the optimizer. One other import distinction is that these runs were hyper-parameter tuned for 1000 update steps, but we increased the experiments to 2000 steps because a few runs seemed to not be fully converging. We believe this lead to some interesting side-effects which we discuss in section 5.1.

**Randomized Hamiltonians:**  Lastly, we run experiments on randomized Hamiltonians. Specifically, we generate each Hamiltonian by combining single qubit measurment terms (randomly sampled Pauli X, Y, or Z gates on random qubits) with 2-qubit measurment terms (tensor products of randomly sampled Pauli X, Y, or Z gates on random qubits). We define the Hamiltonian as

$$\mathcal{H} = \sum_{i=1}^{N_d} c_i(A_i \otimes B_i) + \sum_{i=1}^{N_s} s_i S_i$$

$$c_i, s_i \sim \mathcal{N}(0,\pi) \quad A_i, B_i, S_i \sim \mathcal{U}(\bigcup_{i\in[N]}\{X_i, Y_i, Z_i\}).$$

This benchmark exists for two purposes. First, it is intended to be a problem that is of "medium hardness" that is in-between the 1D Ising and the 2D Heisenburg experiments. Second, by adding randomness not only to the quantum circuit but also the objective, we hope to gain some additional coverage of many possible hamiltonian minimization problems than we did through the prior experiments to see if the insights from them have some evidence of generalization. These experiments are run on systems of 10 qubits. For each run we randmomly sample 10 single qubit terms and 20 2-qubit terms to construct every Hamiltonian. Note that this means, unlike the prior two set of experiments, the energy objective for each optimization procedure differs between run to run. For the random circuit ansatz, we use 30 parameterized single qubit gates and 10 2-qubit gates. Each run is performed for 500 update steps.

## 3.2 Quantum Generative Modelling Experiments

To study a larger variety of quantum learning problems, we also investigate quantum generative modeling. In this setting, instead of trying to produce a state that minimizes and observable, we desire to produce a state that when measured in the full computational basis, matches a provided target distribution. Like with Hamiltonian minimization, we produce our candidate states by using a parameterized circuit. For the same reasons mentioned in section 3.1, we focus on the noise-less setting where we assume we have direct access to the true loss function. In this case we use the negative log-likelihood (NLL) loss.

**Cardinality Constrained Distribution + QCBM:** Our first generative experiment is using the Quantum circuit Born machine [Benedetti et al., 2019] to learn a cardinality-constrained distribution. (So the only randomness in this benchmark is initalization parameters.) The purpose of this experiment is to serve as our baseline for quantum generative results. Because there aren't many optimizer benchmarks for quantum generative modelling, these results on a more standard test model and problem can help us interpret future more heavily randomized results. Specifically we run on a 10 qubit system, with 10 layers of 1 and 2 qubit gates of the QCBM ansatz (illustrated as $L = 20$ in figure 1 of Gili et al. [2023].) The cardinality we constrain to for our distribution is 5, meaning that our target distribution is the uniform distribution over any measurment of all 10 qubits that has 5 1's in the measurment result. Each run is executed for 5000 optimizer steps.

**Randomized Distribution + Random Circuits:** Our next generative experiment is a fully randomized problem. We use the same random layers ansatz used in section 3.1, and our target distribution is fully random. Specifically we use the absolute value of a normal distribution with mean 0 and standard deviation $\pi$, and then divide by the sum to normalize it to a valid probability distribution. In contrast to the other generative modelling experiment, this experiment exists to try to broadly sample many possible generative models and target distributions. For these experiments we run on 5 qubit systems, with 100 parameterized single qubit gates and 30 2-qubit gates in the random layers ansatz. Each run is executed for 5000 optimizer steps.

# 4 Optimizers

In this section we briefly outline all of the optimizers we benchmarked. While this is certainly not a fully exhaustive comparison of all local zeroth-order optimizers, we chose this selection because they cover most methods and heuristics used in SPSA-like methods. Additionally, most of these optimizers have a history of being used for parameterized quantum circuit tasks. Table 1 in the appendix shows the detailed update rules of these optimizers.

**Simultaneous Pertubation Stochastic Approximation (SPSA)** [Spall, 1992] is a commonly used method, both inside and outside the context of optimizing quantum circuits. In a nutshell, SPSA is approximated gradient descent where we randomly sample directions in parameter space. Per step it samples a small random vector from a Rademacher distribution in parameter space, estimates the gradient along that vector with finite difference approximation, and then takes a step along said vector according to the sampled gradient to minimize loss. SPSA is also often used with learning rate and finite difference step size decay, which we also use here.

**AdamSPSA** [Leng et al., 2023] is the application of the Adam optimizer heuristic [Kingma and Ba, 2017] on the SPSA algorithm. Specifically, it estimates via a running sum and updates according to momentum and variance normalization terms.

**2-SPSA** [Spall, 1997] is essentially an approximation of Newton's method, which is gradient descent where the gradient is multiplied by the inverse of the Hessian. To approximate the Hessian, it samples two random vectors (with a Rademacher distribution like in regular SPSA) and evaluates the 2nd order derivative along those two vectors. It then uses a weighted averaging of these samples to provide the Hessian used during optimization. Additionally, because the Hessian estimate can lead to more unstable updates, 2-SPSA also often blocks updates which increase the loss over a certain threshold from the prior value per step.

**Quantum Natural SPSA (QNSPSA)** [Gacon et al., 2021] is a variation of SPSA that utilizes the quantum natural gradient. Similar to classical natural information, using this metric can has a few theoretical advantages to help accelerate and stabilize learning. In practice, this method functions near-identically to 2-SPSA, except that when they sample the metric matrix, they compute the Hessian of the Fubini-Study metric instead of the Hessian of the loss function.

**Guided evolutionary strategies (GES)** [Maheswaranathan et al., 2019] is an evolutionary method with heuristic guiding. However, despite the different name, it is fundamentally very similar to SPSA with only two major differences. First, GES instead samples its random parameter-space vectors with a Gaussian instead of a Rademacher distribution. Second, GES biases the covariance of the sampling Gaussian along the subspace of the recent prior gradients. The intuition behind this choice is that, similar to momentum, that future gradients are more likely that not to be biased in the direction of the most recent prior gradients. However, instead of just increasing the update size in these directions, GES biases the sampling in this direction to increase information gain in this biased direction-of-travel.

**Exponential Natural Evolutionary Strategies (xNES)** [Wierstra et al., 2011] is an extension of evolutionary strategies to improve trainability. In this algorithm, it is assumed we have a multi-variate Gaussian in our space of model parameters, and our goal is to optimize this Gaussian to, in expectation, sample parameters that produce the lowest loss on the underlying problem. This is done by performing stochastic gradient descent on the parameters of the multi-variate Gaussian. xNES then augments this by instead using natural gradient descent to improve convergence guarantees, and utilizes an exponential matrix mapping to make the algorithm more computationally efficient.

**Seperable Natural Evolutionary Strategies (sNES)** [Wierstra et al., 2011] is simply xNES that assumes independence between parameters in order to be even more computationally efficient. It is functionally equivalent to xNES except where the covariance matrix only allows terms along the diagonal.

# 5 Results

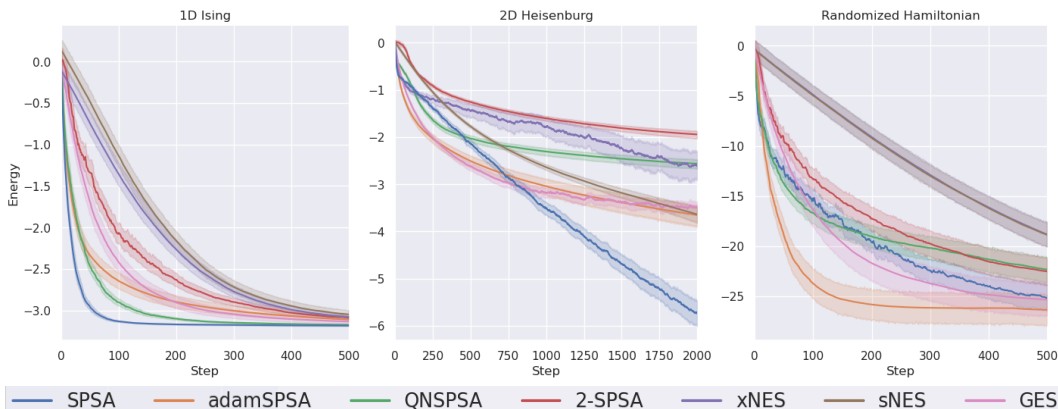

Figure 1: Hamiltonian minimization experiment convergence plots w.r.t. number of optimizer steps. Plots mean value of all runs with the $95\%$ confidence interval. Experiment details can be found in section 3.

We produce two types of plots for all experiments in section 3: convergence plots and box plots. The convergence plots show the average loss of the optimization during each stage of the process, where the colored error area is the $95\%$ confidence interval of the mean.

The box plots illustrate the statistics of the end-result of each optimization run. In these plots, the center line of the colored region is the median loss value. The box region is the interquartile range (the range centered around the median that contains $50\%$ of the samples). The plot whiskers contain

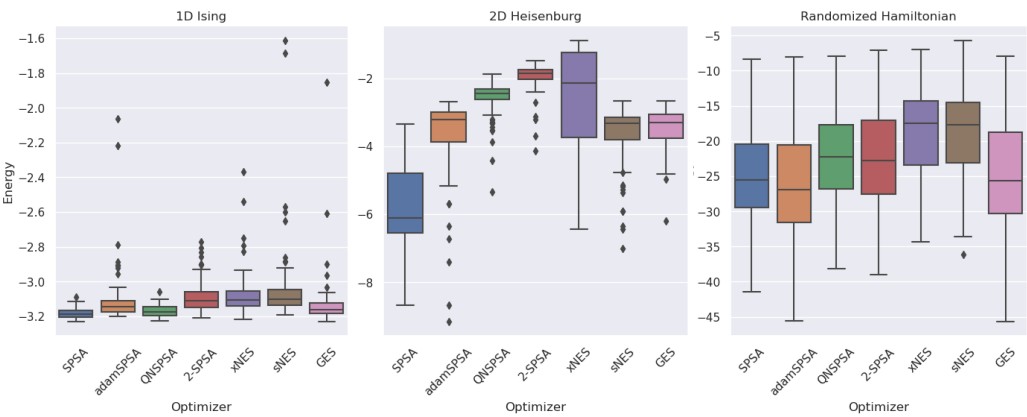

Figure 2: Hamiltonian minimization experiment box plots. Plots the statistics of the final loss value from each run. Experiment details can be found in section 3.

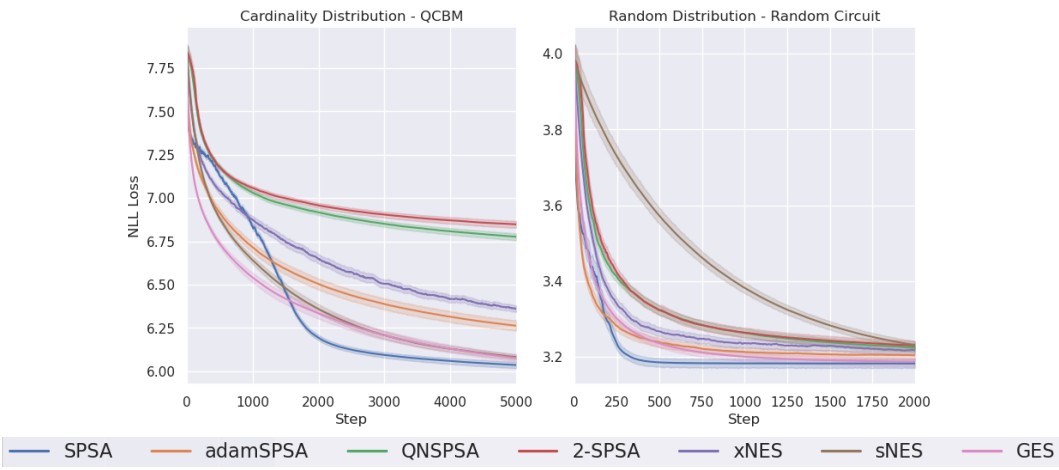

Figure 3: Generative modelling experiments convergence plots w.r.t. number of optimizer steps. Plots mean value of all runs with the $95\%$ confidence interval. Experiment details can be found in section 3.

all points that are within $1.5$ times the size of the interquartile range from the median. All other points are considered outliers and are plotted individually.

The convergence plots are figures 1 and 3 for the Hamiltonian minimization and the generative modelling experiments respectively. The box plots are figures 2 and 4, likewise for the hamiltonian minimization and the generative modelling experiments respectively.

## 5.1 Insights

There are a few main take-aways from these results that we believe these results illustrate. While we don't claim these results are concrete truths, we believe each of them warrant further study.

**1) Hyperparameter tuning for optimizer generalization is extremely important.** Generally this is illustrated by our results showing that most optimizer variants don't show clear benefits compared to their original versions when some hyperparameter tuning is done, which often contrasts the results shown in the original papers. However, this is more specifically illustrated in the convergence plot results of the 2D Heisenburg model shown in figure 1. For this experiment, we hyperparameter tuned to optimizers taking 1000 steps but ran our experimental results out to 2000 steps. If you cut

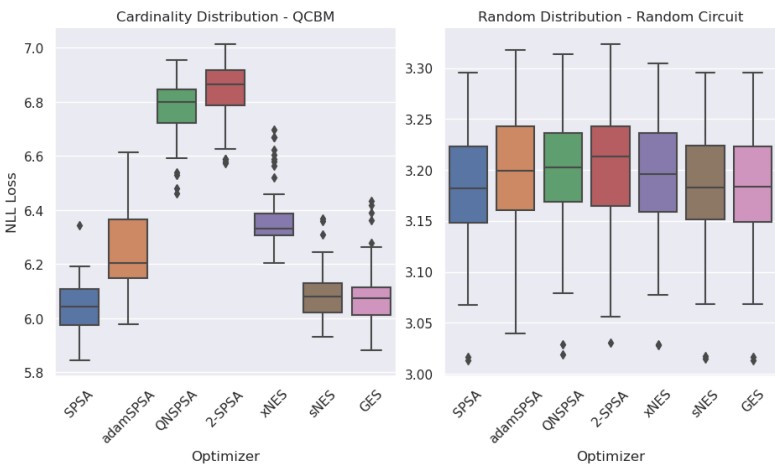

Figure 4: Generative modelling experiments box plots. Plots the statistics of the final loss value from each run. Experiment details can be found in section 3.

this plot off at 1000 steps, it would look very close to the random Hamiltonian experiments where SPSA, GES, and AdamSPSA perform similarly aside from the latter two converging more quickly. However, by choosing to optimize for longer SPSA is suddenly able to do significantly better than all optimizers. While this does raise the question of what a truly realistic hyperparameter tuning scenario is in quantum systems, we think it's likely more fruitful to sidestep this concern altogether and work to design optimizers that are adaptive or more robust to hyperparameter choice.

**2) More elaborate optimization strategies aren't generally better.** In all of these benchmarks, no optimizer clearly out-performs SPSA at the end of the optimization procedure. And the ones that do are often ones like AdamSPSA and GES that rely on relatively simple and cheap heuristics. So while there is certainly something to be said about the theoretical benefits of using a method like QNSPSA, these results indicate that it may be possible to practically achieve better performance for cheaper by using simpler methods like step size decay and guiding heuristics.

**3) When it comes to the convergence speed in the initial phase, you may benefit from acceleration strategies.** It varies depending on the benchmark you look at, but often a few methods are able to converge to lower loss early on in the optimization procedure before being met or overtaken by SPSA. So while it may not be clearly better to use one of these methods as-is, these results indicate it might be possible to develop new adaptive methods to improve the convergence speed of methods like SPSA while not sacrificing overall performance.

# 6 Conclusion

In this work, we benchmark SPSA-like optimizers on a variety of parameterized quantum learning tasks with randomized quantum circuits and randomized objectives. These results provide evidence to suggest that certain heuristics can help accelerate optimization, they often do not perform better than the simpler methods in general. However, we believe that in a broader sense this study helps illustrate the need not only for methods that are adaptive / more robust to hyperparameter choice, but also for broader thought on how we can effectively compare optimizers in quantum systems that aligns with the realistic scenarios they will be used in. While the take-aways from this work are intuitions and ultimately only serve to inform directions of future study, our hope is that this work will inspire more thought into how to best categorize and compare optimizers in quantum learning problems as a whole. (We also include some suggested areas of future studies in section A of the appendix.)

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

# A  Future Works

Beyond the straightforward expansions on this work (larger studies, more optimizers, more relevant randomization, etc.) and works related to the insights in section 5.1, we feel the following would be particularly fruitful follow up studies:

**1) Noise -** This benchmark only compared the noiseless setting, but in practice noise is almost always involved with a quantum computer. Although there are papers that look at shot noise and how to select the right amount of circuit evaluations to still optimize efficiently [Gu et al., 2021], we would like to see a study similar to the one we've done here that adds noise as an additional dimension to study where the transitions of an optimizer performing well v.s. poorly occurs.

**2) Sampling Amount -** For this study, we restricted ourselves to considering only sampling from a minimal number of parameter-difference vectors to get the information needed to take an update step. However, it's possible we could do better by relaxing this constraint. (It's especially surprising methods like GES worked as well as they did in this study, as they were intended to be used with multiple samples per step.) And on the other side of this, can we re-design some aspects of methods that typically benefit from additional samples per step to instead work well in a small-sample setting by aggregating information between steps? Both considerations would be critical for understanding the truly best strategies in this class of optimizer.

**3) Sampling Distribution -** The optimizers we study here use either a Gaussian or a Rademacher distribution to sample parameter vectors. While they both seem to be able to produce effective optimizers, studying in a more principled way the effects of different choices of probability distributions would be interesting. This could especially become more relevant if we look at optimizing parameters that add discrete constraints, as would be the case in some experimental setups, error-corrected quantum computation, coordinate descent parameter-shift rule based optimizers [Schuld et al., 2019], or other more exotic forms of parameterization.

**4) Adaptive Methods -** Along the lines of what was mentioned in section 5.1, studying methods that adaptively change during the optimization process could be fruitful for a number of reasons. First, it could help combine the benefits of multiple strategies. Beyond the optimizers we covered here, there are works that find other ways to accelerate learning. (For instance, Luo et al. [2022] use machine learning to predict optimization trajectories, and Fontana et al. [2023] classically models the loss landscape of certain parameterized circuits.) When it's possible to combine the information from quantum computer queries into multiple methods, having a strategy that can learn to rely either more or less on a specific strategy during parts of the optimization process could allow us to have desirable properties of multiple methods (speed of convergence, ability to optimize well in difficult landscapes, flexibility of a method to work without prior assumptions, robustness to distribution shift, etc.) with a minimal cost-regret overhead. Second, such a method could provide insights into the limits of each of the above methods. By studying in which parts of an optimization process one method begins to be unable to optimize as well as another, it could provide insight to researchers looking to mathematically understand and characterize optimizers and loss landscapes. Lastly, such methods would make future benchmarking studies much simpler. Instead of having to be concerned about what reasonable hyperparameter tuning is and expending the resources to perform it, benchmarks could just compare adaptive versions of the methods in question. (And if said method has a regret bound, they can have precise confidence in the robustness of their results.)

**SPSA**

$$\Delta_i \sim \mathcal{U}(\{-1,1\}^d)$$

$$\epsilon_i = \epsilon_0/i^\gamma, \eta_i = \eta_0/(c+i)^\alpha$$

$$\nabla_{\text{est}}f(\theta_i) = \frac{f(\theta_i + \epsilon_i\Delta_i) - f(\theta_i - \epsilon_i\Delta_i)}{2\epsilon_i}$$

$$\theta_{i+1} = \theta_i - \eta_i\nabla_{\text{est}}f(\theta_i)$$

**AdamSPSA**

$$\Delta_i \sim \mathcal{U}(\{-1,1\}^d)$$

$$\epsilon_i = \epsilon_0/i^\gamma, \eta_i = \eta_0/(c+i)^\alpha, \beta_i = \beta_0/i^\lambda$$

$$\nabla_{\text{est}}f(\theta_i) = \frac{f(\theta_i + \epsilon_i\Delta_i) - f(\theta_i - \epsilon_i\Delta_i)}{2\epsilon_i}$$

$$\left(m_1 = \nabla_{\text{est}}f(\theta_1), v_1 = (\nabla_{\text{est}}f(\theta_1))^2\right)$$

$$m_i = \beta_i m_{i-1} + (1-\beta_i)\nabla_{\text{est}}f(\theta_i)$$

$$v_i = \gamma v_{i-1} + (1-\gamma)(\nabla_{\text{est}}f(\theta_i))^2$$

$$\theta_{i+1} = \theta_i - \frac{\eta_i}{\sqrt{v_i}+\delta}m_i$$

**GES**

$$U_i = \text{orthonormal basis of span of}$$
$$\{\nabla_{\text{est}}f(\theta_{i-k}), \ldots, \nabla_{\text{est}}f(\theta_i)\}$$

$$\Sigma_i = \frac{\alpha}{n}I + \frac{1-\alpha}{k}UU^T, \Sigma_{0,\ldots,k} = \frac{1}{n}I$$

$$\Delta_i \sim \mathcal{N}(0, \sigma^2\Sigma_i)$$

$$\nabla_{\text{est}}f(\theta_i) = \beta\frac{f(\theta_i + \Delta_i) - f(\theta_i - \Delta_i)}{2\sigma^2}$$

$$\theta_{i+1} = \theta_i - \eta\nabla_{\text{est}}f(\theta_i)$$

**2-SPSA**

$$\Delta_i, \Delta_i' \sim \mathcal{U}(\{-1,1\}^d)$$

$$\nabla_{\text{est}}f(\theta_i) = \frac{f(\theta_i + \epsilon_i\Delta_i) - f(\theta_i - \epsilon_i\Delta_i)}{2\epsilon_i}$$

$$\delta f = f(\theta_i + \epsilon\Delta_i + \epsilon\Delta_i') - f(\theta_i + \epsilon\Delta_i)$$
$$- f(\theta_i - \epsilon\Delta_i + \epsilon\Delta_i') + f(\theta_i - \epsilon\Delta_i)$$

$$\hat{H}_i = \frac{\delta f}{2\epsilon^2}\frac{\Delta_i(\Delta_i')^T + \Delta_i'(\Delta_i)^T}{2}$$

$$H_i = \frac{i}{i+1}H_{i-1} + \frac{1}{i+1}\hat{H}_i$$

$$\theta_{i+1} = \theta_i - \eta H_i^{-1}\nabla_{\text{est}}f(\theta_i)$$

**xNES**

$$\Delta_i, \Delta_i' \sim \mathcal{N}(0,I), B_1 = I$$

$$z = \theta_i + \sigma_i B_i^T\Delta_i, z' = \theta_i + \sigma_i B_i^T\Delta_i'$$

$$u = 0.5 \text{ if } f(z) < f(z') \text{ else } -0.5$$

$$u' = 0.5 \text{ if } f(z') < f(z) \text{ else } -0.5$$

$$\nabla_\mu J = us + u's'$$

$$\nabla_M J = u(ss^T - I) + u'(s's'^T - I)$$

$$\nabla_\sigma J = \text{tr}(\nabla_M J)/d$$

$$\nabla_B J = \nabla_M J - \nabla_\sigma J \cdot I$$

$$\sigma_{i+1} = \sigma_i\exp(\eta_\sigma/2 \cdot \nabla_\sigma J)$$

$$B_{i+1} = B_i\exp(\eta_B/2 \cdot \nabla_B J)$$

$$\theta_{i+1} = \theta_i + \eta_\mu\sigma_i B_i\nabla_\mu J$$

**QNSPSA**

$$\Delta_i, \Delta_i' \sim \mathcal{U}(\{-1,1\}^d)$$

$$F(\theta,\theta') = |\langle\psi(\theta)|\psi(\theta')\rangle|^2$$

$$\nabla_{\text{est}}f(\theta_i) = \frac{f(\theta_i + \epsilon_i\Delta_i) - f(\theta_i - \epsilon_i\Delta_i)}{2\epsilon_i}$$

$$\delta F = F(\theta_i, \theta_i + \epsilon\Delta_i + \epsilon\Delta_i')$$
$$- F(\theta_i, \theta_i + \epsilon\Delta_i)$$
$$- F(\theta_i, \theta_i - \epsilon\Delta_i + \epsilon\Delta_i')$$
$$+ F(\theta_i, \theta_i - \epsilon\Delta_i)$$

$$\hat{H}_i = \frac{-\delta F}{4\epsilon^2}\frac{\Delta_i(\Delta_i')^T + \Delta_i'(\Delta_i)^T}{2}$$

$$H_i = \frac{i}{i+1}H_{i-1} + \frac{1}{i+1}\hat{H}_i$$

$$\theta_{i+1} = \theta_i - \eta H_i^{-1}\nabla_{\text{est}}f(\theta_i)$$

**sNES**

$$\Delta_i, \Delta_i' \sim \mathcal{N}(0,I)$$

$$z = \theta_i + \sigma_i\Delta_i, z' = \theta_i + \sigma_i\Delta_i'$$

$$u = 0.5 \text{ if } f(z) < f(z') \text{ else } -0.5$$

$$u' = 0.5 \text{ if } f(z') < f(z) \text{ else } -0.5$$

$$\nabla_\mu J = us + u's'$$

$$\nabla_\sigma J = u(s^2 - 1) + u'(s'^2 - 1)$$

$$\sigma_{i+1} = \sigma_i\exp(\eta_\sigma/2 \cdot \nabla_\sigma J)$$

$$\theta_{i+1} = \theta_i + \eta_\mu\sigma_i B_i\nabla_\mu J$$

Table 1: Optimizer algorithms. Illustrates a step of each optimizer in equation form. Note that these algorithms may slightly differ from the original works due to simplifying choices / constraints we made in this study. $f$ is the loss function and $\theta$ are the parameters per step. $\psi$ is the parameterized quantum circuit model used by $f$. $\mathcal{U}$ is the uniform distribution and $\mathcal{N}$ is the Gaussian distribution. $(\cdot)^2, \sqrt{(\cdot)}$ are element-wise on vectors. All other un-defined variables are hyperparameters.

