# OpenReview forum: "Randomized Benchmarking of Local Zeroth-Order Optimizers for Variational Quantum Systems"
_NeurIPS.cc/2023/Workshop/AI4Science — NeurIPS2023-AI4Science Poster_

### Official Review · Reviewer_xWYR · 2023-10-24
**A very timely study on the importance of optimizers in variational quantum systems**

**Rating:** 7
**Confidence:** 4

**Review:**

I appreciate the thorough experimentation and findings from the authors' experiments.

I have a question about the randomized Hamiltonians. Are these Hamiltonians natural, i.e. are there any physical systems form which randomized Hamiltonians is a good model? Elaborating on the choice of these testbeds would be helpful. Thanks!

---

### Meta-Review · Area_Chair_jnNj · 2023-10-27

**Recommendation:** Accept (Poster)
**Confidence:** 3

**Metareview:**

I personally the recommendation for this paper to be hard.

This work provides an overview and detailed study of the effectiveness of SPSA-like optimizers on various parameterized quantum learning tasks with randomized quantum circuits and objectives.
While there's not much novelty within the paper (sounds more like a survey paper), I also find the connection to Machine Learning somewhat loose. The content of the paper is interesting from a physics and quantum computing standpoint, although I am not entirely sure whether it fully satisfies all criteria for this workshop.

However, taking into account the input from referee xWYR, the fact that I find the paper interesting, and that the workshop is prone to accept rather than reject, I recommend the acceptance of this work.